# Effects of the Family Solidarity on Romanian Left behind Children

**DOI:** 10.3390/ijerph19105820

**Published:** 2022-05-10

**Authors:** Aniela Matei, Elen-Silvana Bobârnat

**Affiliations:** 1National Scientific Research Institute for Labour and Social Protection—INCSMPS, 010643 Bucharest, Romania; silvana.bobarnat@incsmps.ro; 2Faculty of Sociology and Social Work, University of Bucharest, 030018 Bucharest, Romania

**Keywords:** family solidarity, migration, children, caregiving, parenting

## Abstract

In families that have at least one parent working outside the country’s borders, known in the literature as transnational families, family solidarity is undergoing changes. The aim of this paper is to explore the process of transforming family solidarity in the context of migration and to identify the effects of the family solidarity on children left behind. A qualitative research methodology was employed consisting in 24 in-depth interviews with parents and grandparents from transnational families, in the two Romanian regions with the highest number of children left behind and with high poverty rates. Our results show that the decrease of material and financial support provided from the parent left abroad has great implications for the feeling of unity with the family and for the material and emotional well-being of children. The risk factors for the children are a lack of financial support, which translates to material deprivation and creates the context for school dropout, lack of emotional support, and poor closeness between the child and departed parent, which relate to a disrupting emotional experience for the children. Findings provide new insights in capturing the relationship between family solidarity and the well-being of the child.

## 1. Introduction

The mobility of people between the states of the European Union for work is an objective of the European Community and is addressed, with varying degrees of generality, in the founding treaties of the Union, in its regulations, directives, and decisions [1,2]. As the difficulties encountered by the transnational family, especially minor members and their dependents, have been identified, Community legislation has been enriched and adapted (Directive 2004/38/EC).Thus, the European Union is constantly improving the legislative context for maintaining the unity in transnational families defined in the literature [3,4] as a new family model with the following characteristics: geographical dispersion of a family caused by migration, maintenance of the relations between the family members across borders, and the construction of a collective welfare and unity, named by Bryceson and Vuorela as ‘familyhood’, even across national borders. In accordance with existing definitions in this paper, we define transnational families as families that live some or most of the time separated from each other, yet hold together and create something that can be seen as a feeling of collective welfare and unity, namely familyhood, even across national borders. The issue of social protection for workers and their families is addressed at the legislative level by the European Union in Regulation (EEC) no. 1408/71 of the Council of June 1971, amended by Regulation no. 1247/1992. Directive 2004/38/EC on the right of citizens of the European Union and their family members to reside freely within the territory of the Member States, which specifies the Union’s desire to maintain family unity by granting the right to move and reside freely within the territory of its countries and family members.

Technological innovation, which allows communication and travel at distances that until recently were unimaginable, enhances the opportunities for maintaining the unity and solidarity in families with members left abroad. Despite the opportunities to improve their living standard and maintain the connectiveness between family members, some of the Romanian children that are left at home face economic, social, and emotional risks [5,6,7].

Due to the freedom of movement within European Union and economic disparities between countries, labour force migration has increased in recent years. It is estimated that in Romania, Bulgaria, and Poland, altogether 500,000–1 million children are affected (Council of Europe, (London, UK) 2020). The scientific literature captures a number of positive effects of the migration process for work outside the borders of the country of origin, but also identifies a wide range of negative effects [8,9,10].

Free movement of labor force after 2000 has long been considered a factor in ensuring competitiveness in the European Union [11,12]. Since Romania’s entry into the European Union, the number of migrant workers has steadily increased, but the recent economic crisis has slowed the emigration process as the circular and return economy has intensified [13].

In the present research, we look at the transformations within the families in the context of migration abroad and their effects on children left behind. In the first part of the article, we will analyze the studies that have captured the consequences of migration for family solidarity. Then, based on the results of a qualitative research methodology (24 in- depth interviews with parents and grandparents from transnational families), we will explore the process of transforming family solidarity in the context of migration and identify the effects of the family solidarity on left behind children.

## 2. Background

The scientific literature on the process of emigration of Romanians addresses a variety of topics, from the economic and social effects of emigration, remittances, demographic profiling of migrants, to brain drain and circular and return migration etc. [14]. If in the first years after the fall of communism the migration process was determined by ethnic reasons [15], empirical studies developed after 2010 identified a number of determinants of the emigration process of Romanians referring to high poverty rates in some regions of Romania, low wages, poor education and health systems in Romania, etc. [16,17,18,19,20]. Scientific literature [21,22] indicates that migrants go to other countries in search of a better standard of living for themselves and their relatives who have remained in the country.

Considered a positive effect of migration, remittances, defined as financial transfers that compensate for the loss of human capital and brain drain due to emigration [23], have contributed to economic development [24], but only in the short term. In the long run, remittances did not ensure economic growth, but stimulated the final consumption of households and income inequality [25], negatively affecting the competitiveness of exports and the motivation to work.

International migration is a global phenomenon that influenced the family structure in such an amount that generated the appearance of a new type of family, the transnational family, whose members live in different countries but share the feeling of unity [4].

Previously, other societal changes reflected on the family, and they could be a source of inspiration in understanding the current transformations of it. Beaujout and Ravanera in 2008 identified the transition from organic solidarity within the family using a collaborative model, consisting in both organic and mechanic solidarity, and a demographic transition, consisting in both the reduction of family size due primarily to the decrease in the number of children, and the flexibility in the conjugal relationship that has increased [26] (pp. 76–78).

In an attempt to understand the effects of migration on the family solidarity in transnational families with children left in the origin country, our first research question is *How did family solidarity transformed after the migration abroad of the parent(s) in these families?* The factors of family solidarity identified in the literature are: immigration history and geographical proximity [27], geographical distance having impacts on the frequency of on family reunification situations, the lack of face-to-face interaction with the family in the country of origin, but also the presence of children [26] (p.88).

Roberts et al., in 1991 [28], and later Bengtson and Oyama in 2007 [29], systematized the theories regarding family solidarity and identified six dimensions of the concept: 1. *associational solidarity*, referring to the interaction between family members, under the aspects of frequency and patterns of interaction, 2. *consensual solidarity*, referring to the agreement between family members regarding values, attitudes and beliefs, 3. *functional solidarity*, consisting in the exchange of services and assistance between family members, 4. *affectual solidarity*, referring to the sentiments and evaluations that the family members express about the relationships between them, 5. *normative solidarity*, regarding the obligations incorporated in child and parental statuses, as well as norms regarding the importance of family values, and 6. *intergenerational family solidarity/structural solidarity*, referring to the number of the family members, the type of kinship between them, and their geographical proximity. Other authors define family solidarity as “the feeling of loneness with the family” [30].

Referring to the families in the Romanian society, Voinea [31] (pp. 37–38) mentions the importance of the cohesion and marital solidarity, defined as convergence of the action of all members of the group in achieving a common goal, namely, normal development, harmonization of group life. Also referring to the contemporary Romanian families, Popescu mentions the importance of the family norm consisting in the solidarity between conjugal couple’s partners [32] (p. 73, p. 178) and she defines solidarity as “mutual trust and support, mutual respect and understanding, fidelity” [32] (p. 58, p. 73).

The different ways of defining the family solidarity, either by its six dimensions, or by the general feeling that the family members share, led to our second research question: *How do the different aspects of family solidarity relate to one another?*

When studying the children left behind, a series of research [33,34,35,36] identifies the aspects of the migration experience, e.g., remittance level, predictability/frequency of remittances, migration of mother vs. father, and duration and frequency of migration and communication, which generate health, economical educational, and psycho-social outcomes affecting children. The causes and effects are mediated by variables, such as characteristics of the children (age, gender, ability), characteristics of the household (size and structure, education levels, labour capacity, social support, income, attitudes, bargaining power), and societal characteristics (livelihood options, societal values, service provision, social protection, framework). These findings lead us to the third research question: *What are the effects of the family solidarity on children left behind?*

The theoretical lens of family solidarity in the context of migration has been used to study the relation between adults and children that migrated abroad and their elderly parents left in the origin country [37,38,39], some of them referring to Romanian society [40]. However, the relation between the family solidarity in transnational families and its effects on children is understudied both in Romania and abroad. This motivated us to look at the migration effects on children left at home from the perspective of family solidarity theory.

## 3. Materials and Methods

### 3.1. Research Objectives

A qualitative research approach was used for this study. Through the present research we aim to answer three research questions. *RQ1: How did family solidarity transformed after the migration abroad of the parent(s)? RQ2: How do the different aspects of family solidarity relate to one another? RQ3: What are the effects of the family solidarity on children left behind?*

We define family solidarity both as an (1) *overall direct and subjective evaluation of it*, and *by looking at (2) two theoretical axes of the concept*. We ask respondents about the feeling of unity with the family [41] and the feeling that they share the same goals [31] and how those feelings changed after the migration abroad. We also ask respondents about two aspects of family solidarity, material support (both financial and consisting on services such as care, cleaning, cooking, helping with homework etc.) and emotional support (consisting in contact and interaction between family members), but we encouraged respondents to talk about the changes in family life related to the children that took place after the migration, in order to identify other aspects of family solidarity.

### 3.2. Data Source

The data source used in this paper is represented by 24 qualitative interviews carried out between 6 July and 13 July 2021. Interview guide was developed and applied by the authors within a research project on the topic of transnational families carried out by the National Scientific Research Institute for Labour and Social Protection (INCSMPS). The use of qualitative methods in migration research is recommended in order to address topics of interest for contemporary migration themes [42].

An in-depth interview guide was designed and used to address the following topics of discussion: (1) the role of socializing agent, (2) the role of emotional support, (3) the role of income provider, (4) family solidarity, and (5) strategies to maintain family solidarity. For this article, the discussion topics related to *family solidarity* and *strategies to maintain family solidarity* were retained for analysis. The questions addressed to the interviewees referred to: family unity after the migration process, breaking family ties, moments of crisis in the family after the migration of the parent, convergence of objectives of transnational family members, children’s problems, what the family gained, and it what lost as a result of the parent’s departure abroad.

The interviews were conducted face-to-face with parents/grandparents from families who have children up to 17 years old and who have a parent (or both parents) who went to work outside Romania (12 interviews in the North-East Region of Romania and 12 interviews in the South-East Region of Romania).

The selection of development regions was made based on the analysis of indicators related to poverty and temporary and permanent migration, and we selected the two regions of Romania most affected by poverty. At the end of 2020, data recorded by the National Authority for the Protection of the Rights of the Child and Adoption in Romania indicated that the phenomenon of transnational families was more pronounced in the case of families where one parent went to work abroad (52,474 with one parent left for compared to 13,253 children with both parents working outside the country). The theoretical saturation model was used in the process of collecting data from this qualitative research.

### 3.3. Sampling and Selection Criteria

The selection of development regions was made based on the analysis of indicators related to poverty and temporary and permanent migration, and we selected the two regions of Romania most affected by poverty. A specific sampling technique in qualitative research was used, namely the purposive sampling technique, suitable for choosing respondents for qualitative studies using the interviews as a data collection tool [43]. Participants in the current study were chosen using the following criteria: (i) the quality of the respondent in relation to the child: parent/grandfather (10 of the respondents were grandparents and 14 were parents); (ii) at least 6 months experience of migration for parent/parents (under 3 years of migration experience/time since the partner/children’s parent(s) went abroad: 4 participants; between 3 and 5 years of migration experience/time since the partner/children’s parent(s) went abroad: 13 participants; 6 or more years of migration experience/time since the partner/children’s parent(s) went abroad: 6 participants); (iii) environment of residence: urban/rural in order to reflect the structure of the phenomenon, mainly specific in Romania for rural and small towns (9 interviews were conducted in urban areas and 15 interviews were conducted in rural areas), (iv) age of children: to cover as much as possible situations from the earliest ages up to the maximum age limit of 17 years (the average age of the child: 10 years; percentage of children above the average age: 38.7%; percentage of children under the average age: 51.6%).

### 3.4. Qualitative Data Collection Processes and Analysis Approaches

The survey based on in-depth interviews was chosen as the appropriate research method because it can produce a rich understanding [44] of the relation between the family solidarity in transnational families and its effects on children, this relation being understudied in Romania.

The 24 interviews (12 in each of the two regions of economic development of Romania) were conducted based on the interview guide developed by the research team. Interviews were audio recorded and all ethical issues regarding confidentiality, informed consent and anonymity were addressed during the process of data collection. The interviews were conducted face-to-face at the home of the respondents, between 6 July and 13 July 2021 with parents/grandparents from families who have children up to 17 years old and who have a parent (or both parents) who went to work outside Romania (12 interviews in the North-East Region of Romania and 12 interviews in the South-East Region of Romania). The average length of an interview was 30 min.

The research team used the qualitative research software NVivo12 Pro to manage the data (Table 1). The content analysis technique was used for data analysis [45]. The data were structured in the following categories: (1) the subjective evaluation of the family solidarity; (2) the dimensions of the family solidarity: material support and emotional support and (3) changes in children emotional estate and behavioural estate. Last, we compared (using Crosstab and Matrix Coding queries in NVivo12 Pro) different degrees of family solidarity, different ways of fulfilling the financial and the material support, and changes in the emotional and behavioural states of children.

## 4. Results

### 4.1. The Factor of Parent(s) Migration

To understand the engine of the migration and the expectations of the families’ members in this transformative context, we first asked our interviewees about the reason to migrate. For all interviewed participants (N:24), the factor of migration was an economic one, the desire to ensure the *downward financial support* for the family members. Prior to the migration, the parents were unable, in different degrees, to provide financial support for their children, even though sometimes they were helped by the extended family members (grandparents). These parents were struggling to make ends meet. The unfulfilled material needs consisted of food for children, clothing and footwear, providing an appropriate shelter, ensuring decent living conditions and utilities, and supplies for school.


*“[Previous to the migration the parents] were working [informally] in the village and they were paid by day of work …*
*10 lei [Romanian currency; 2 Euro], 50 lei [Romanian currency; 5 Euro] … they starved to death in the house and quarrelled with the women.”*
(Grandmother, NE Region, 7 grandchildren)


*“[Previous to the migration the parents and their children] … stayed with me for 5 years. So we stayed in a two-room apartment, me and my husband and they… It was very hard.”*
(Grandmother, NE Region, 1 grandchild)

### 4.2. Post-Migration Family Solidarity

Interviewees were asked about the evolvement of the post-migration family solidarity both from a direct and subjective perspective, regarding the feeling of unity with the family [30], and from an indirect perspective consisting in the material and emotional support within the families [28,29]. Then, we looked at how these two perspectives relate to each other.

#### 4.2.1. Direct and Subjective Evaluation of the Family Solidarity

When looking at the feeling of unity with the family, we identified two relevant stages on the evolution of the family solidarity: *on the short-run* and *on the long-run* transformations.

On the short-run, at the moment of the departure and in the first weeks after the migration, the interviewees either (1) felt that their families were at risk of disintegration, or (2) they felt relieved because the migrating parent left.

In the first category were the majority of the interviewees in this study, who described the transformations that their families faced in the context of migration in the same terms of insecurity. Whether the relationship between the migrant parent and the child carer is a marital or filial relationship, migration is a critical moment for family solidarity.


*“When they left [I felt that the family was no longer united], when the children began to cry, because they missed them.”*
(Grandmother, NE Region, 2 grandchildren) 


*“[When my sun goes abroad] it’s like he become a stranger. When he leaves, he leaves and it’s as if when he comes [back home] we don’t know each other anymore … when you meet again we need a while … we take it [our relationship] from the beginning … when we get closer to each other, he leaves again.”*
(Grandmother, NE Region, 1 grandson)


*“I was sad, [I feel] alone [after my husband left abroad] … but we are moving forward.”*
(Mother, NE Region, 2 children)

The second category consisted in the families in which the migrating parent was presenting aggressive behaviour, and his migration was a chance for the family members to live in harmony.


*“It’s not how it should be [the family unit] … [he was calling me every 5 min … [and now, after the migration, it’s] the same. [when he was home] I couldn’t stand the situation any more … He got drunk, he beat me.”*
(Mother, SE region, 6 children).

However, with the years passing, the family solidarity evolved differently for the studied families. When analyzing the feeling of unity with the family on the long-run, two categories of cases emerged: those who at least maintained if not improved their family unity after migration (*Families in solidarity*) and those who experienced a deterioration of the family solidarity (*Families in risk of family disintegration*).

*Families in solidarity* described that even if at the moment of the migration they were insecure because of how the family will manage to stick together, over time their family solidarity was at least maintained at the pre-migration level, if not even improved, making them feel closer to the family members left abroad.


*“… we felt more united. He decided to make a sacrifice for us, we made a sacrifice for him … we stayed at home and we didn’t make any problems and … we felt more united… The material part was what we won after migration. And I think the closeness between me and my husband, as family… Between us as husbands. I think the trust between as has also become more [strong]…”*
(Mother, NE Region, 1 child).

The second category, *Families at risk of family disintegration,* slowly lost the feeling of unity with the migrating members of the family.

We further analysed the factors of family solidarity on the long-run as well as the effects on children.

#### 4.2.2. Indirect Evaluation of the Family Solidarity

Based on the previous literature [28,29], we identified two thematical axes of the concept *family solidarity*: material support and emotional support.

When analyzing the data, the importance of the material support for the feeling of unity with the family emerged. The material support does not consist only in the financial provision, but also has other components: the financial provision, the norm referring to the consistent effort to provide downward financial support (the “doing your best for providing downward financial support” norm), the services provision, and the norm of reciprocity within the family (Table 2).

The *families in solidarity*, that experienced family unity enhancement or at least its maintenance, also mentioned the fulfilment of the financial aims that determined the migration or the consistent effort of the migrant parent to meet the financial objectives of the migration.

*“We are satisfied that we have fulfilled everything we had to accomplish financially”*.(Mother, SE Region, 1 child)

Comparatively, in the *families in risk of disintegration*, that faced the deterioration of the feeling of unity, the interviewed members not only mentioned that the migrant parent disengaged from providing financial support, but they expressed distrust regarding the efforts that the migrant parent made in order to accomplish his/her duty as financial provider.


*“At the beginning there was a period when he sent money and a package, after a while they became scarce… he motivated not to earn as at the beginning… because he also has expenses.”*
(Mother, SE Region, 1 child)

The *families in solidarity* mentioned the provision of services within the family, such as: helping the children with homework through ICT mediated communication (material support offered to the child), indirect supervision of child school participation through consistent discussions with the teachers (material support offered to the child carer), a form of support related to the feeling of unity between the family members.

When the migrant parent does his best to provide financial support, the accomplishment of a reciprocity family norm by the family members remaining in the country strongly enhances family unity.


*“He decided to make a sacrifice for us, we made a sacrifice for him… we stayed at home and we didn’t make any problems and … we felt more united…”*
(Mother, NE Region, 1 child)

Emotional support also emerged as important for the feeling of unity with the family, and the concept of emotional support includes: the frequency of indirect interactions between the migrating parent(s) and the family members left at home (through ICT), the frequency of direct interactions (returns at home or visits abroad), and the quality of interactions (Table 3).

The analysis of the interviews indicates that not all the families in which the migrant parent accomplished financial duties mentioned that they also maintained the feeling of being a family. This led us to the hypothesis/idea that financial support is not a sufficient condition for the family unity. The families satisfied with the financial support provided by the migrant parent that contrary to this support are *at risk of disintegration*, mentioned the scarcity of direct contact between family members (rare returns at home of the migrant parent), especially on important moments for the members of the family.


*“Not much [I still have the feeling of family]. When he comes home, I forget… I still call my children… He comes very late and rarely…”*
(Mother, SE Region, 2 children)

For some families, the lack of financial support (including the norm consisting in the parent making consequent effort to financially support the children) is doubled by the lack of emotional support, a combination of aspects that profoundly affects the feeling of family unity.


*“When the first Christmas passed and we didn’t see each other, when C.’s first birthday passed and we didn’t see each other [I felt that family members have different objectives]… I thought that things would repeat themselves indefinitely and that I had to find the strength to see this child that she manages on her own, not only through my powers.”*
(Grandmother, SE Region, 1 child)

The *families in solidarity* talked about frequent and prolonged indirect (through ITC mediated communication) and direct contact between the child and the migrating parents (returns at home or visits abroad of the members felt at home), and they also talked about the quality of interaction (the parent and the child making trips together, the child sharing important aspects of his life with the migrating parent). Where this was the case, the infrequent direct interaction between the child and the migrating parent seems to be compensated by the quality of the interaction (e.g., the child prevalently shares important aspect of his life with the migrating parent through ICT mediated communication).

At this point of the analysis, we can say that both material and emotional support are important for the feeling of unity with the family.

### 4.3. The Effects of Weak Family Solidarity on Children

In this section, we will concentrate on the effects of the financial provision and the post-migration quality of interaction between the children and the migrating parent on children well-being.

The data highlighted four types of negative effects of the lack of emotional and material support on children: material deprivation, prolonged unpleasant emotional experience, school-absenteeism, risky behaviour (e.g., self-harm).

When the migrating parent cannot provide the financial support (either because his/she disengages from the material support role, or he/she earns a small amount of money), the children are at risk of material deprivation (e.g., lack of food, heating on winter). The two variables, financial support and material deprivation, are mediated by: (1) the financial support provided by the child carer, which in some cases is enough to protect the children from the material deprivation, by (2) the material support offered by NGOs, churches, etc. that generally provide clothes and food, but can only attenuate the material deprivation, and by (3) the financial support provided by the children themselves, an aspect that generates the risk of school absenteeism. School absenteeism also manifested in cases in which the migrating parent is the constant income provider, but prior to as well as after the migration if they cannot provide enough money for family material objectives in the absence of the income generated by the children (see Table 4).

The type of emotional experience of the children after the process of migration depends, among other variables, on the quality of interaction between the children and the migrating parent (see Table 5). Generally, post-migration, the children face disturbing emotional experiences (sadness, nervousness, loneliness, introversion, shyness), but when prior to the migration the parent had a poor quality of interaction with the children (e.g., manifested physical and verbal violence), the migration of the abusing parent generates a positive emotional experience for the children (see Table A1).

In the cases in which the children face disturbing emotional experience after migration, the degree and extent of the disrupting emotional experiences of the child depends on multiple aspects of family solidarity, some of them involving members of the extended family. When the children (1) *continue to receive emotional support from the migrating parent(s)*, under all aspects, including frequent ICT-mediated interactions, frequent encounters especially in a strongly socially regulated context with a good quality of interactions, (2) with *emotional support from the child carer*, (3) *with whom the child has a close emotional relationship and spends a long time with prior to the migration*, the disturbing emotional experiences faced are at a small degree and reduced in duration (see Table A2). However, when the migrating parent(s) disengage(s) from the emotional support, under the aspects previously discussed, and where the child and the child carer do not have a close previous relationship, the disruptive emotional experience becomes deep and perpetuates in time. Children that lack both material and emotional support experience extreme emotional experiences, followed by risky actions like self-harming behaviours (see Table 4, Table 5 and Table A3).

## 5. Discussions

A number of studies in the field identify migration as a moment of crisis for family solidarity in transnational families [26,27,28,29]. Our analysis based on qualitative interviews confirms this aspect for Romanian transnational families, indicating that no matter if the relationship between the migrant parent and the left behind partner/member of the family (grandfather of the child) is a marital or filial one, migration is a critical moment for family solidarity. Some of the families faced infidelity between members of the marital couple, a component of the solidarity between conjugal couple’s partners [32]. In this case, the dissolution of the solidarity inside the conjugal couple was doubled by the impossibility to provide appropriate financial support for the children.

The data collected indicate that the provision of the financial support for the members of the family left behind, the accomplishment of the family norm regarding the consistent effort that a parent must make to provide financial support for their children, even when they cannot fulfil this duty, the provision of services inside the family by the migrant parent(s) other than financial support, and the reciprocity family norm are all important factors for family solidarity.

The literature emphasizes that the emotional support provided to children from transnational families is affected after the migration [3,4]. Our analysis based on qualitative interviews indicates that the provision of financial support is not a sufficient condition for maintaining the solidarity of the family as an essential condition for the good development of the children. The reduced frequency of contact between family members (rare returns on home of the migrant parent), especially at important moments for the members of the family negatively influences family solidarity by making the child vulnerable. For some families, the lack of financial support (including the norm consisting in the parent making consequent effort to financially support the children) is doubled by the lack of emotional support, a combination of aspects that profoundly affects the feeling of family unity.

The present research highlights the importance of both material and emotional support for the children well-being. Remittances are very important for improving the living standards of the left behind children [17,18,19,20], but equally important is the emotional support for the children psychological well-being. Strong family solidarity can help reduce the negative effects on the emotional state of children whose parents have gone to work abroad. In fact, there are studies [6] that have identified a correlation between the migration of both parents or the migration of the mother and children experiencing depression symptoms. The desire to improve the material well-being of the children motivates the parents to migrate abroad and the most mentioned positive effects of migration are the enhancement of the financial status of the family [5,6].

The lack of financial support leads to material deprivation of the children and creates the risk of school absenteeism. The risk of school absenteeism was identified by other authors, too [6,7]. However, in the mentioned research, the school drop-out was connected more to the lack of surveillance and parents’ authority, while our research highlights that the migration does not necessarily protect the children from the material deprivation, as the financial burdens, generated by structural factors related to the inefficiency of welfare policies, encourage the children to miss school and start working on the informal labour market. Material deprivation has long standing effects on children’s health, cognitive performance, and emotional and behavioral status [46], so public interventions targeting the alleviation of the material deprivation of minors are necessary.

The lack of emotional support translates into a disruptive emotional experience for the children, an experience that perpetuates in time and that is accompanied by risky actions, such as auto-destructive behaviors. The importance of emotional support for the children’s psychological well-being is consistent with the results of other studies [6]. In these cases, public interventions consisting both in specialized emotional support for the children and in parenting guidance for the parents are required.

Further research could focus on identifying and testing other factors of family solidarity and child well-being in the context of migration (e.g., specific prescriptions of the children role related to financial provision for him and the family) or on answering the research questions on a larger scale through quantitative research.

The results have both a theoretical role in capturing the relationship between solidarity and the well-being of the child, as well as a practical role in capturing effective strategies of family solidarity.

## 6. Conclusions

This study used a qualitative research methodology to better understand the process of transforming family solidarity in the context of parental migration for work outside the country with direct implications on the material and emotional well-being of children. The results showed that financial support and emotional support are related to the subjective and direct evaluation of the family solidarity, with families lacking the financial provision and emotional connection with the migrating parent also mentioning a weak family unity. The deterioration of the feeling of unity is amplified by the disengagement of the parent(s) from both material support and emotional support, the second consisting in a low frequency of returns to the country of the migrant parent and his absence in the strongly regulated social moments (holidays, anniversaries, etc.).

Our study adds to the existing frameworks of qualitative research results that emphasize the factors which influence the migration process outside the borders of the country, the transformations in the material and emotional aspects of family solidarity during the migration process, their influence on the feeling of unity with the family, and the effects on the well-being of children.

Despite the limitations of this study (geographical coverage restricted to only two development regions of Romania), the results of this qualitative research can help practitioners in the field and public policy makers to better understand the relationship between family solidarity and the well-being of the child, in presenting concrete strategies for maintaining solidarity and ensuring the well-being of the child. The results also support the need to change the intervention policies in Romania in order to solve some structural issues that cause the migration of parents for work outside the country, mainly related to employment (wages) and family welfare policies, such as low-paying jobs of the parents and possession of low skills jobs.

## Figures and Tables

**Table 1 ijerph-19-05820-t001:** Phases of data analysis.

Analysis	Means
Data mining/data coding of transcripts of face-to-face in-depth interviews data (*n* = 24)	NVivo 12 Pro
Thematic analysis of face-to-face in-depth interviews data (*n* = 24)	NVivo 12 Pro

**Table 2 ijerph-19-05820-t002:** The material support and the feeling of unity with the family.

Analysis Categories	No. of Family with Improved or Maintained Solidarity Status	No. of Family with Impaired Solidarity Status
**The financial provision**		
Accomplished	19	0
Not accomplished	1	4
**The “doing your best for providing downward financial support” norm** (only those who did not accomplished financial support were included)		
Mentioned as accomplished	0	4
Not mention as accomplished	1	0
**The services provision**		
Provision of other services within the family	2	0
No provision of other services within the family	18	4
**The norm of reciprocity within the family**		
Mentioned as accomplished	2	0
Not mention as accomplished	18	4

Source: The Matrix Coding outputs generated using NVivo 12 Pro.

**Table 3 ijerph-19-05820-t003:** The emotional support and the feeling of unity with the family.

Categories	No. of Family with Improved or Maintained Solidarity Status *(Families in Solidarity)*	No. of Family with Impaired Solidarity Status *(Families in Risk of Disintegration)*
**The frequency of indirect interactions (through ICT)**		
Multiple times a week	16	1
Once a week or less	1	3
Unclear	3	0
**The frequency of direct interactions (returns at home or visits abroad)**		
At least once a year	7	1
Less than once a year	1	3
Unclear	11	0
Not the case (left from less than 1 year)	1	0
**The quality of interactions with the migrating parent(s)_post-migration**		
Good	19	2
Poor	1	2

Source: The Matrix Coding outputs generated using NVivo 12Pro.

**Table 4 ijerph-19-05820-t004:** The effects of weak financial provision on children well-being.

Analysis Categories	Financial Provision Accomplished (no. of Families)	Financial Provision Not Accomplished (no. of Families)
**Material deprivation**		
Material deprivation_No	19	3
Material deprivation_Yes	0	2
**Post-migration emotional experience of the children**		
Short- term unpleasant (minutes, hours, days) emotional experience	15	3
Long-term or permanent unpleasant emotional experience	0	2
No changes	3	0
Pleasant emotional experience	1	0
**Risky behaviours**		
Risky behaviours (e.g., self-harming)_No	19	4
Risky behaviours (e.g., self-harming)_Yes	0	1
**School-absenteeism**		
School_absenteeism_No	17	3
School_absenteeism_Yes	2	2

Source: The Matrix Coding outputs generated using NVivo 12 Plus.

**Table 5 ijerph-19-05820-t005:** The effects of poor quality of interaction with the migrating parent(s) on children well-being.

Analysis Categories	Quality of Interaction_Post Migration_Good (no. of Families)	Quality of Interaction_Post Migration_Poor (no. of Families)
**Post-migration emotional experience of the children**		
Short- term unpleasant (minutes, hours, days) emotional experience	18	0
Long-term or permanent unpleasant emotional experience	0	2
No changes	3	0
Pleasant emotional experience	0	1
**Risky behaviours**		
Risky behaviours (e.g., self-harming)_No	21	2
Risky behaviours (e.g., self-harming)_Yes	0	1
**School-absenteeism**		
School_absenteeism_No	19	1
School_absenteeism_Yes	2	2

Source: The Matrix Coding outputs generated using NVivo 12 Pro.

## Data Availability

Data sharing is not applicable to this article.

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
