# Peer review of "Effects of the Family Solidarity on Romanian Left behind Children"

_ijerph, 2022, doi:10.3390/ijerph19105820_

Round 1

Reviewer 1 Report

  1. Line #14, misspelled “in-depth”.
  2. Line #53, what does "…leaving a better life…" mean?
  3. Line #56, need a comma between "countries" and "labour", making the sentence understandable.
  4. Line #73-#74, what does "...had in view of high poverty rates…" mean?
  5. Line #102-#113, need to clearly identify or specify the 6 dimensions, such as assigning a number preceding each of them.
  6. Line #261, what made you believe that those are social-desirable answers, not genuine answers?
  7. Line #271-#272, why is this a genuine answer, not a social-desirable answer?
  8. Line #305-#307, are these described activities done by the migrant parent? Also, what is ICT?
  9. Line #326-329, the description is the same as described in line #276-#279.
  10. Line #336, what constitutes a quality interaction?
  11. Line #354, what are "…closes and food"?
  12. Line #380 and #434, what is "…auto-destructive behaviours"?
  13. Line #427-#428, the description “…financial burdens encouraging the children to miss school and start working on the informal labour market” would also happen if the migrant parents did not go aboard but continued low-pay jobs and suffered difficult financial situation at home town. The child would eventually work and miss school to support the family. Hence, the issues are low-paying jobs, possession of low skills, and welfare policies of the country. In other words, these are structural issues, not necessarily individual choices.
  14. Line #455, misspelled “weak”.
  15. In Conclusions section, the authors will need to add implications for intervention and policies.

Author Response

Dear Reviewer,

Dear Reviewer,

Thank you very much for your comments. We treated them with track changes function in the manuscript.

We look forward to hearing from you,

Aniela Matei

Elen-Silvana Bobarnat

Reviewer 2 Report

Abstract: The abstract is too long. It should be reduced by adding the aim of the paper, the used method, and the achieved results.

Introduction: The aim of the paper should be clarified. The sub-section “1.2. Consequences of migration for family solidarity” should be moved to the background section. Moreover, at the end of the section, the structure of the paper should be provided.

Background: A background section should be added. Some of the content of the sub-section 1.2. “Consequences of migration for family solidarity” can be used. More recent studies should be added.

Materials and Methods: The content and structure of the interviews are not clear. A more detailed description should be added.

Results: This section should be improved a lot. A more detailed description of the obtained results should be added. Moreover, I suggest to summerise the list of single answers provided by some respondents in a table followed by a discussion.

Conclusion: Limitations of the study and future studies should be added.

Author Response

Dear Reviewer,

Thank you very much for your comments. We treated them with track changes function in the manuscript.

We look forward to hearing from you,

Aniela Matei

Elen-Silvana Bobarnat

Round 2

Reviewer 2 Report

The authors satisfied all the comments; an English spell check is required